# Comparative Description and Analysis of Oyster Aquaculture in Selected Atlantic Regions: Production, Market Dynamics, and Consumption Patterns

Johannes A. Iitembu [1,*], Daniel Fitzgerald [2,†], Themistoklis Altintzoglou [3], Pierre Boudry [4], Peter Britz [5], Carrie J. Byron [6], Daniel Delago [7], Sophie Girard [4], Colin Hannon [8], Marcia Kafensztok [9], Francisco Lagreze [10], Jefferson Francisco Alves Legat [11], Angela Puchnick Legat [11], Adriane K. Michaelis [12], Ingelinn Eskildsen Pleym [3], Simone Sühnel [9,13], William Walton [12] and Åsa Strand [14,*]

1. Department of Fisheries and Ocean Sciences, University of Namibia, Henties Bay 13005, Namibia
2. Ecole Supérieure d'Agricultures, 55 Rue Rabelais, 49000 Angers, France; daniel@smallaxepeppers.com
3. Nofima, Muninbakken 9-13, 9291 Tromsø, Norway; themis.altintzoglou@nofima.no (T.A.); ingelinn.eskildsen.pleym@nofima.no (I.E.P.)
4. Département Ressources Biologiques et Environnement, Ifremer, 29280 Plouzané, France; pierre.boudry@ifremer.fr (P.B.); sophie.girard@ifremer.fr (S.G.)
5. Department of Ichthyology and Fisheries Science, Rhodes University, Makhanda 6140, South Africa; p.britz@ru.ac.za
6. School of Marine and Environmental Programs, University of New England, 11 Hills Beach Road, Biddeford, ME 04005, USA; cbyron@une.edu
7. Ocean Era Inc., 73-970 Makako Bay Drive, Kailua-Kona, HI 96740, USA; dan@ocean-era.com
8. Marine and Freshwater Research Centre, Atlantic Technological University, Dublin Road, H91 T8NW Galway, Ireland; colin.hannon@atu.ie
9. Primar Aquacultura, Estrada RN 03, km 10, Sítio São Félix, Piau, Tibau do Sul 59178-000, RN, Brazil; primarorganica@gmail.com (M.K.); ssuhnel@gmail.com (S.S.)
10. Center of Marine Research, Federal University of Paraná, Av. Beira-Mar, s/n, Pontal do Sul, Pontal do Paraná 83255-976, PR, Brazil; lagreze@ufpr.br
11. Brazilian Agricultural Research Corporation, Embrapa Coastal Tablelands, Av. Beira Mar 3250, Aracaju 49025-040, SE, Brazil; jefferson.legat@embrapa.br (J.F.A.L.); angela.legat@embrapa.br (A.P.L.)
12. Virginia Institute of Marine Science, College of William and Mary, Andrews Hall 440, Gloucester Point, VA 23062, USA; amichaelis@vims.edu (A.K.M.); walton@vims.edu (W.W.)
13. AquaInspiration and Innovation, Rua Túlio de Oliveira, 195, Armação, Florianópolis 88066-303, SC, Brazil
14. Department of Environmental Intelligence, IVL Swedish Environmental Research Institute, Kristineberg 566, 45178 Fiskebäckskil, Sweden
* Correspondence: jiitembu@unam.na (J.A.I.); asa.strand@ivl.se (Å.S.)
† Current address: Small Axe Peppers, New York, NY 10001, USA.

**Abstract:** In the face of an increasing world population and a subsequent need for an increase in sustainable and healthy food production, low trophic species, such as oysters, emerge as a promising alternative. However, regional variations in oyster production techniques, market dynamics, and consumption patterns create challenges for both the global and local industry's growth. In this study, a descriptive qualitative analysis of oyster markets across seven Atlantic regions was carried out. The Pacific oyster (*Crassostrea gigas*) was found to be farmed in most Atlantic regions except the US but is classified as invasive in Sweden and potentially invasive in South Africa. Other farmed and/or harvested species include native species (*C. gasar and C. rhizophorae*) in Brazil, the American cupped oyster (*C. virginica*) in the US, and the European flat oyster (*Ostrea edulis*) in France, Sweden, and the US. In Irish farms, Pacific oysters are primarily for export to European markets. The marine aquaculture sectors of Sweden, South Africa, and Namibia, as well as Brazil's farming for *C. gasar*, were found to be underdeveloped. This study also observed a variation in licensing, property rights, and regulatory frameworks. Financial challenges for small businesses, ecological implications of seed production techniques, biosecurity risks, and public health considerations are emphasized as critical areas for attention. This study offers valuable insights into the selected markets and can serve as a useful resource for policymakers, aquaculture practitioners, and stakeholders in optimizing global shellfish industry strategies.

**Keywords:** Atlantic regions; oysters; production; markets; aquaculture; low trophic

**Key Contribution:** Qualitative analysis of oyster markets in seven Atlantic regions highlights variations in the species farmed, production techniques, and market dynamics. Key challenges identified include financial hurdles for small businesses, ecological concerns related to seed production, biosecurity risks, and public health issues. These findings could inform policymakers and stakeholders in refining oyster industry growth strategies at both global and local levels.

## 1. Introduction

Humans have consumed shellfish since ancient times. Archaeological studies provide evidence of its consumption dating back to the initial migrations of *Homo sapiens* out of Africa, approximately 100,000 to 200,000 years ago [1]. Oysters, among shellfish species, have been prized as aphrodisiacs from the time of Roman emperors to contemporary Casanovas. For millennia, oysters have sustained civilizations from the Ancient Romans [2] to various Chinese dynasties [3] up to modern consumers. Few types of seafood can boast such a storied history. This popularity has resulted in overexploitation globally [4]. In Europe, overfishing led to the depletion of European flat oyster (*Ostrea edulis*) beds starting from the 18th century [5–7]. In a similar vein, eastern American oyster (Crassostrea virginica) reefs in the United States experienced a decline by the 1940s [6]. Globally, oysters represent the leading farmed molluscan species in terms of quantity produced, with increasing production since 1950 [8]. Global oyster production is dominated by China, while the next five countries with significant oyster aquaculture production include France, the US, South Korea, Japan, and the Philippines [8]. Asia dominates the global production and consumption of oysters [9] and production in other regions has also seen an increase since 1950 [8].

Oysters also play significant ecological roles in addition to their importance as a food source. They are recognized as ecosystem engineers due to their direct or indirect control over resource availability for other organisms [10]. Oysters trap organic-enriched particles with their shells and create three-dimensional structures, thereby facilitating the presence of additional fauna and seaweed, particularly in soft-sediment environments [11,12]. As filter feeders, oysters contribute to water clarity through their filtration process. This increased clarity allows for greater light penetration, which benefits adjacent vegetated habitats such as seagrass beds [13]. These vegetated habitats are vital as nursery areas and for carbon sequestration [13]. Furthermore, oyster filtration helps reduce the occurrence of harmful algal blooms, which have significant ecological and economic impacts [13,14].

As food, oysters are a highly nutritious source, containing essential nutrients not provided by land-based proteins [15,16]. They are considered one of the most nutrient-dense seafood options [17] while offering a high-protein, low-fat product that contains good levels of healthy $\Omega$-3 fatty acids [9]. Oyster aquaculture, recognized as low-trophic, is also valued for its reduced environmental impacts, lower feed requirements [18], and potential significant contributions to circular economics [19]. Various research initiatives have focused on expanding the products and processes of low-trophic species in marine aquaculture value chains across the Atlantic [20,21]. However, globally, oyster production has faced recurring setbacks due to disease and parasitic outbreaks [8]. Furthermore, translocations of farmed oysters between geographical areas and the introduction of new species for culture have contributed to the dispersal of invasive species, pathogens, and parasites over time [22].

Despite these challenges, the global production of oysters has shown an upward trend, indicating a sustained increase in oyster aquaculture production for various markets [8]. In terms of regional and country/region-specific productions, it has stagnated in most countries and the increase is primarily driven by the Asian market [8]. While oyster production and consumption have been increasing in some regions [8,23], there are also areas

where oysters are underutilized as a resource [24]. The underutilization of oysters in some regions may be attributed to the marketing of oysters, which is influenced by factors such as growth, mortality, quality, price [25], and existing sanitation programs (food safety) [26]. Additionally, there is a lack of knowledge about the comparative characteristics of various regions regarding production, consumption patterns, and market dynamics. This study, therefore, aimed to conduct a qualitative description of oyster production, consumption patterns, and market dynamics in seven regions (the US, Brazil, France, Sweden, Ireland, South Africa, and Namibia) along the Atlantic Ocean, to enhance understanding of these markets. The research questions included understanding how oyster production methods differ among the seven Atlantic regions, the implications of these differences for the industry's sustainability and growth, the unique consumption patterns of oysters in each region, and how market dynamics and strategies vary across these regions. Comparing these similarities and differences among the regions can serve as a crucial resource for understanding and optimizing global and local oyster production, marketing, and consumption strategies. Additionally, the findings from this study can also be used as a basis for detailed investigations of oyster farming strategies and practical adoption of farming methods in specific markets.

## 2. Methods and Materials

### 2.1. Study Areas

The geographical regions included in this study encompassed North America, South America, Europe, and Africa, all of which border the North and South Atlantic Oceans (Figure 1). These regions were selected based on their participation in or association with the AquaVitae project, to provide a snapshot of variability among nations/regions based on overall market trends. Due to the large geographical scale and diversity of the Brazilian and American markets, these areas were further subdivided into northern US, southern US (including the Gulf of Mexico), northeast Brazil, and southern Brazil to facilitate more detailed analyses of these regions (Figure 1). The chosen markets within these regions included (a) markets with high levels of production and consumption (mature): France, the US, and southern Brazil; (b) markets known for low levels of oyster consumption and production (immature): Sweden and northern Brazil; and (c) markets that did not comfortably fit into either category (a) or (b) (mixed): Namibia, South Africa, and Ireland.

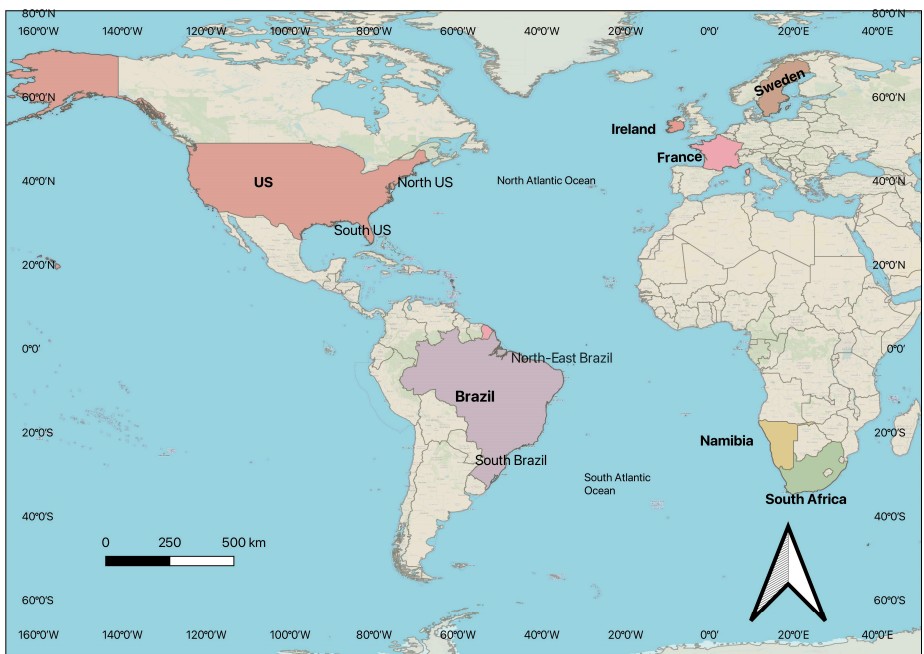

**Figure 1.** A world map illustrating the geographical regions (shaded in colours) examined in this study.

## 2.2. Data Collection and Analysis

This study employed questionnaires (open-ended questions) and personal communications with academic experts from the selected markets (Table 1) and literature searches. The experts were selected based on their substantial knowledge of the selected markets and their participation in or association with the AquaVitae project. The purpose was to conduct a qualitative description of the selected markets to gain insights into the historical trends in oyster aquaculture, consumer perceptions of oyster aquaculture, and current market trends related to oyster consumption and production. Qualitative descriptive analysis was used because this study drew from the perspectives of the selected experts, employed a purposeful selection of experts, and involved content analysis of various literature and reports [27,28]. The questionnaires, given to the experts, included a current overview of oyster production and consumption in each market. The data extracted from the questionnaires and expert interviews were utilized to profile the selected markets using the PEST (political, economic, social, and technological) analysis framework to provide insights into market characteristics [29]. For the application of PEST, the political factors (P) account for the political situation. In this study, they included governance, licensing, and regulatory measures related to health considerations. The economic factors (E) cover issues that may influence business operations and profitability. This study included issues related to the challenges and opportunities for financing. Social (S) factors cover aspects related to consumers and producers, with this study incorporating aspects of the roles of producer organizations and cooperatives. The technological (T) factors cover issues related to operations and production, which are critical for the long-term future of oyster farming in various markets. Technological considerations in this study included seed production techniques and regulation, the role of hatcheries in oyster seed production, the advantages of triploid oysters, production techniques, postharvest processing, the importance of depuration facilities, and food safety through traceability. The data used included production numbers, production methods, aquaculture cultivation, wild harvest, native and exotic species, import and export markets, consumer profiles, safety standards, and value-added products relevant to the chosen markets. Other parameters for each market considered included licensing structures, governance, food safety requirements, financing/investment availability, insurance availability, hatcheries/seed production, triploid production, water monitoring programs, producer cooperatives and organizations, and native oyster cultivation/restoration (Table 2).

**Table 1.** List of experts selected from each market considered.

| Market | Expert ID | Affiliation |
|---|---|---|
| US (north) | 1 | University of New England |
| US (north) | 2 | University of New England |
| US (north) | 3 | University of New England |
| US (south) | 4 | Virginia Institute of Marine Science |
| US (south) | 5 | Virginia Institute of Marine Science |
| Brazil (north) | 6 | Brazilian Agricultural Research Corporation (EMBRAPA) |
| Brazil (north) | 7 | Primar Aquacultura |
| Brazil (south and north) | 8 | Primar Aquacultura, AquaInspiration and Innovation |
| Brazil (south and north) | 9 | Universidade Federal do Paraná |
| France | 10 | Ifremer |
| France | 11 | Ifremer |
| Sweden | 12 | IVL Swedish Environmental Research Institute |
| South Africa | 13 | Rhodes University |
| Namibia | 14 | University of Namibia |

**Table 2.** Summary of factors from PEST (political, economic, social, technological) analysis framework and market characterization.

| | Northern US | Southern US | Northern Brazil | Southern Brazil | France | Ireland | Sweden | South Africa | Namibia |
|---|---|---|---|---|---|---|---|---|---|
| **Market characterization** | Mature | Mature | Immature | Mature | Mature | Mixed | Immature | Mixed | Mixed |
| **Species** | *Crassostrea virginica* | *C. virginica* | *Crassostrea gasar, C. rhizophorae,* and *C. gigas* | *Crassostrea gasar, C. rhizophorae,* and *C. gigas* | *C. gigas* and *O. edulis)* | *C. gigas* and *O. edulis)* | *C. gigas* and *O. edulis* | *C. gigas* and *Striostrea margaritacea)* | *C. gigas* |
| **Fishery category** | Aquaculture (A) and capture fishery (C) | Aquaculture (A) and capture fishery (C) | Aquaculture and capture fishery | Aquaculture | Aquaculture | Aquaculture | Aquaculture and capture fishery | Aquaculture and capture fishery | Aquaculture |
| **Production statistics (2019)** (Aquaculture = A and capture fishery = C) | 148,000 t (A) and 38,000 t (C) | | 1700 t (A) and 1000 t (C) | | 86,000 t(A) | 10,700 t (A) | 21 t (A) | 383 t (A) and 23 t (A) | 350 t (A) |
| **Political (P), Economic(E), Social (S), and Technological (T): PEST considerations** | | | | | | | | | |
| Leasing of seabed/licensing (P) | x | x | | x | x | x | x | x | x |
| Oyster sanctuaries/protected areas (P) | x | x | | | | x | x | | |
| Economic incentives (loans, tax breaks) (E) | x | x | | x | x | | | x | x |
| Disaster relief/crop insurance (E) | x | x | | | x | | | | x |
| Efficient and timely permitting process (P) | x | x | x | x | x | | | x | x |
| Extensive seed production (wild seed collection/pond production) (T) | | x | x | | x | x | x | | x |
| Hatchery seed production (T) | x | x | x | x | x | x | x | | x |
| Use of triploids (T) | x | x | | x | x | x | | | |
| Producer cooperative (S) | x | x | x | x | x | x | x | x | x |

**Table 2.** *Cont.*

| | Northern US | Southern US | Northern Brazil | Southern Brazil | France | Ireland | Sweden | South Africa | Namibia |
|---|---|---|---|---|---|---|---|---|---|
| **Political (P), Economic(E), Social (S), and Technological (T): PEST considerations** | | | | | | | | | |
| Water monitoring programs (P) | x | x | | x | x | x | x | x | x |
| Food safety regulation (P) | x | x | x | x | x | x | x | x | x |
| Postprocessing facilities (T) | x | x | x | x | x | x | x | x | x |
| Traceability (T) | x | x | | x | x | x | x | x | x |

## 3. Results and Discussions

### 3.1. General Description of Production, Market, and Consumption Patterns of the Selected Markets

3.1.1. The United States

The native *C. virginica* is the primary oyster species on the eastern seaboard of the United States and is the most valuable marine aquaculture species in the US [30]. In 2019, Atlantic Ocean oyster landings, including aquaculture and wild harvest, were valued at approximately USD 192 million, with over 148,000 metric tons of farmed oysters and almost 38,000 metric tons of wild-harvested oysters (Table 2) [31]. Various production techniques are employed for oyster production, ranging from harvesting wild oysters from wild grounds to on-bottom culture using shell deployment to collect naturally produced seed, or young oysters transplanted to private, leased grounds. More intensive methods used involve off-bottom, floating, and other types of container culture, with oyster seed obtained from commercial hatcheries. Equipment commonly used varies by location, largely depending on local system dynamics (e.g., water depth, tidal fluctuation, and accessibility) and US state regulations. In the US, nearshore aquaculture adheres to federal regulations and is managed at the state level. In some cases, states allow for even more localized management of oyster aquaculture, and it is managed and permitted at township or county levels.

In terms of markets, wild oysters are typically sold at lower prices compared to aquaculture-farmed oysters and have traditionally been associated with the canning market. However, there is a shift in consumer preferences towards fresh, raw oysters. In 1970, 76% of oysters produced were consumed as fresh and raw, increasing to 92% by 1994 [32]. Shellfish consumers, according to national seafood surveys, tend to have higher education and household income than average [33,34]. They are also more likely to be over 50 years of age [35]. Experienced oyster consumers show a preference for farmed oysters, while infrequent oyster consumers are more inclined towards the concept of wild-caught oysters [36].

Certification under the National Shellfish Sanitation Program (NSSP) is mandatory for all oyster dealers in the US to market oyster products across state lines. However, the proportion of oyster production exported is relatively small, partly due to the 10-year ban on shellfish trade with the EU, although this was recently lifted [37]. This shift may be linked to changes in the typical oyster consumer. Additionally, the US imports oysters from both China and South Korea, including both farmed and wild species [38].

3.1.2. Brazil

Oyster production in Brazil primarily revolves around the *Crassostrea* genus, which includes two native species (*Crassostrea gasar* (=*Crassostrea brasiliana*) and *C. rhizophorae*) and one exotic species (*C. gigas*, introduced in Brazil in 1974), with the latter being the most produced and consumed. In 2019, Brazil's total oyster production reached 2700 tons, consisting of 1700 tons from aquaculture and 1000 tons from capture fisheries, with a farm oyster value estimated at USD 1.8 million (Table 2) [31]. The native species (*C. gasar* and *C. rhizophorae*) are widely distributed across Brazilian estuaries and have been harvested by low-income fishing communities in various mangrove areas [39]. However, in several regions, oyster growers expressed concerns over the low market value of wild oysters, which imposes limitations on both market prices and farmed oysters. Sometimes, juveniles are harvested from the mangrove and used for grow-out in fisheries-based aquaculture systems. Some producers also employ seed collectors to capture native species of *Crassostrea* ssp. seeds. Oyster hatcheries play a crucial role in producing seeds for *C. gigas* farming in southern Brazil, where no wild seed collection occurs. In 2019, the estimated production of hatchery seeds reached about 46,000 seeds, although there are limited hatcheries for the native species [40], with the seed of *C. gasar* being produced in a private hatchery in the northeast of Brazil and in a public hatchery in southern Brazil.

Regarding production techniques, the majority of producers utilize suspended lanterns in longline systems and bags for grow-out. In the southern regions, suspended systems, such as longlines with lantern nets (off-bottom culture) are common, particularly in shallow and protected areas with minimal tidal variation [40], with one case in on-bottom culture in Paraná. In contrast, bottom-based systems, particularly fixed tables with plastic bags, are prevalent in other states, especially in the northeast and north. There are also cases of pond-based production of oysters in the northeast. The variation in production techniques across states is primarily influenced by the specific grow-out areas. For instance, in Santa Catarina and Rio de Janeiro, cultivation occurs in bay areas near the coast, while in the northeastern and northern states, estuarine areas are primarily utilized for oyster grow-out, with Paraná and São Paulo in both conditions, bay and estuarine areas. The duration of the grow-out cycle of *C. gigas* and *C. gasar* ranges from 6 to 12 months, depending on the harvest size and area of production.

The consumption of native oysters from wild capture is a longstanding tradition in all coastal states of Brazil, although it lacks comprehensive documentation. On the other hand, the consumption of *C. gigas* oysters, which is well documented, is based on aquaculture production. The majority of oyster products are consumed locally by Brazilians and tourists [41], particularly during the summer months (December to February), when the majority of the population goes on vacation and there is an increase in the flow of tourists to coastal regions. In the northeast region, there is also an increase in consumption in the month of July, as it is also a vacation period. The Brazilian oyster market exhibits segmentation between high-end and low-end markets. Cultured oysters sold in restaurants or stores are typically associated with the high-end market, while oysters collected by artisanal fishermen and sold in public seafood markets or by street vendors on the beaches cater to the low-end market. Oyster products in Brazil can be marketed as whole, freshly shucked, steam-cooked, boiled, or frozen [39]. In terms of regulatory measures, as of 2023, only one state in southern Brazil, Santa Catarina, has implemented a monitoring program for oyster production, with Geographical Indication (GI) for *C. gigas* from Florianópolis. Additionally, Alagoas in the northeast region of Brazil possess a government-certified shellfish depuration facility. Besides initiatives to export oysters (such as to Singapore, the US, and China) there are no exports from, or imports to, the Brazilian oyster market yet.

3.1.3. France

France holds the position of the largest oyster producer in Europe. In 2019, the total oyster production amounted to approximately 86,000 tons (Table 2) [31], with the Pacific oyster (*C. gigas*) accounting for 93% and the native European flat oyster (*O. edulis*) comprising only 1165 tons (7%) [22]. The value of oyster cultivation in the same year reached USD 446 million, with the Pacific oyster contributing USD 438 million and the flat oyster generating USD 8 million [22]. The production of flat oysters remains limited due to the presence of parasites of the genus *Bonamia* and *Marteilia* [7]. While wild harvest plays a minimal role in production, the cultivation of farmed oysters involves the traditional seasonal collection of oyster seeds during summer. Oyster hatcheries of Pacific oyster (*C. gigas*) and European flat oyster (*O. edulis*) now meet approximately 30–50% of the seed demand, supplying seed almost year-round. Notably, 90% of the hatchery-produced seed consists of triploid oysters [42]. In areas where local seed collection is feasible, many oyster farmers utilize both sea-collected and hatchery-produced seeds.

The primary production method in France involves the use of racks and trestles located in the intertidal zone. Oysters are cultivated from the seashore to deeper waters, primarily in protected shallow bays at depths ranging from 10 to 15 m [42]. In the Mediterranean Sea, where the tidal range is limited, longline suspended culture is predominantly employed [42]. Deep-water culture (on-bottom) is practiced in south Brittany, where the seed is deployed in large concessions and subsequently harvested through dredging. The duration of the grow-out cycle ranges from 2 to 4 years, depending on the specific area, applied techniques, oyster species, and the type of seed used. The transfer of developing oysters between

production areas is a common practice to leverage the advantages offered by different regions, such as seed collection, on-growing, and finishing processes.

Oysters have traditionally been considered a luxury product, primarily sold live and consumed raw [7,43]. Oyster consumers tend to be older than average and mainly belong to higher income brackets [7]. Consumption patterns involve at-home consumption, which is highly seasonal, and sales through super/hypermarkets [7]. In France, oyster production follows the European Union Food Hygiene Regulations, which establish standards for end products. However, Marennes-Oléron oysters stand out with a Protected Geographical Indication (PGI), certifying the refinement of Pacific oysters in "claires" (or ponds originally used to produce sea salt) within the Marennes-Oléron basin. This certification ensures the production of the highly prized fine-de-claire oysters, known for their exceptional quality [44]. France's oyster exports remain relatively low, primarily directed towards Italy, China, Hong Kong, the Netherlands, and Spain. Import-wise, France receives just over 5100 tons of oysters from Ireland, primarily as juveniles to finish growing in French waters.

### 3.1.4. Ireland

Ireland's total oyster production reached approximately 10,700 tons (2019) (Table 2) [31], with the majority consisting of Pacific oysters (*C. gigas*) at 10,460 tons and flat oysters (*O. edulis*) at 256 tons. The total value amounted to USD 51 million for Pacific oysters and nearly USD 1.4 million for flat oysters [31]. Oyster production is spread across 11 counties in Ireland, with Waterford and Donegal collectively accounting for around 60% of the country's production by tonnage [45]. While Ireland has several hatcheries, the industry heavily relies on imported seeds primarily from hatcheries in France and, to a lesser extent, from British hatcheries [46].

The main cultivation method for Pacific oysters in Ireland is through the use of bags and trestles. Ireland's oyster industry is primarily export-focused, with France accounting for 88% of Ireland's exports between 2012 and 2014, with 78.4% destined for the French market in 2021 with the increase in purchasing from other European markets. In 2021, the output volume and sales value of 10,624 tons valued at EUR 47.55 million (USD 51.8 million) [47]. The industry has been supported by French–Irish joint ventures [7]. Ireland also imports 241 tons of oysters, mainly seeds from the UK, and 72 tons of oysters from France. Oyster production follows the European Union Food Hygiene Regulations, which establish standards for end products. *O. edulis* production capacity is growing in Ireland with the redevelopment of spatting ponds by aquaculture enterprises and the further redevelopment of native oyster production beds at sea by fishermen's co-ops through the deployment of cultch and coupelles [48].

### 3.1.5. Sweden

Sweden's oyster production is relatively small, with a total of 21 tons (Table 2) [31]. Aquaculture production in Sweden primarily focuses on the native flat oyster (*O. edulis).* The Pacific oyster (*C. gigas)* has been invasive in Sweden since its establishment in 2006 [49], and its cultivation is prohibited. For wild harvest, approximately half of the oyster catch consists of Pacific oysters while the other half comprises European flat oysters with a small portion of farmed *O. edulis*. The aquaculture production is constrained by seed availability and seeds are sourced mainly using sea-based collectors. A hatchery established in 2006 produces insufficient seed for sale to the rest of the industry [50]. The harvest of wild *O. edulis* (fisheries) remains relatively stable over time. In contrast, the harvest of wild *C. gigas* has seen an increase in recent years, with production rising from 1 ton to 8 tons between 2009 and 2019.

Wild oysters are harvested through diving (*O. edulis* and *C. gigas*) or handpicking (*C. gigas*) [51]. Farming of *O. edulis* is carried out using surface-based floating cages or suspended cages from longline systems or other structures. Sea-based seed capture is the primary method for oyster farming. Oyster consumption patterns in Sweden align with Scandinavian per capita consumption of shellfish, which remains relatively small

compared to many other developed countries in Europe and North America [52]. The oyster market is primarily local, with oysters mainly consumed in restaurants and as half-shells. Grilled and gratinated wild-harvested Pacific oysters are gaining popularity. However, the demand from Swedish consumers exceeds the supply, resulting in a steady increase in oyster consumption. Imported oysters, mainly from the Netherlands and France, meet the demand, while there are no exports from Sweden. Import volume has increased significantly, from 30 tons in 2002 to 350 tons in 2010 [52]. The native flat oyster commands a premium price compared to locally harvested Pacific oysters, which are priced higher than imported Pacific oysters.

3.1.6. South Africa

Aquaculture in South Africa's marine environment is underdeveloped due to the limited presence of estuaries or sheltered bays suitable for culture operations. In 2019, the total oyster production in South Africa was 406 tons, with 383 tons from aquaculture, mainly the Pacific oyster (*C. gigas)*, and 23 tons from native rock oyster (*Striostrea margaritacea*) captured for subsistence consumption and commercial use (Table 2) [53]. The aquaculture value amounted to USD 5 million [31]. The introduction of the Pacific oyster to South Africa occurred in 1973, while the current market for native oysters relies entirely on wild harvesting [54]. Wild oysters are either collected by subsistence fishers in the Eastern Cape province for their own consumption or are sold for income [55], or by licensed commercial harvesters in the southern Cape for sale to the restaurant trade. Early efforts to farm native rock oysters were abandoned due to their slower growth compared to Pacific oysters and limited cultivation knowledge [53]. Commercial oyster culture in South Africa is predominantly located in the Western Cape, specifically in the Saldanha Bay area, where the Pacific oyster is farmed in baskets suspended on longlines [53]. Currently, it takes three to four months to grow oysters to a marketable size, with the industry relying on imported oyster seed from Chile, England, and local hatcheries [53].

The South African oyster market is a niche market primarily supplying restaurants, with higher sales during December to March. South African oysters are mainly sold fresh (80–85%), with the wild rocky oyster sought after for its rich, wild taste. It commands a price premium over farmed Pacific oysters, with cultured oysters and rock oysters fetching a 20–30% premium. Oysters are sold shucked, whole frozen, or frozen in half shells, with a small portion value-added with sauce and crumbs. Smoked and canned oyster products in South Africa are imported, as well as the majority of frozen oyster products, as imports dominate the postharvest value-added market [56]. Exporting oysters from South Africa faces challenges due to the lack of international health certification standards, limiting access to most countries. Currently, oysters are exported to southeast Asian countries like Hong Kong, where health standards are less stringent. Efforts are underway to implement a shellfish water quality monitoring program to obtain EU certification and expand market access. However, high production costs and low volumes tend to make South African oyster producers uncompetitive in export markets [57].

3.1.7. Namibia

In 2019, Namibia's oyster production reached 350 tons, exclusively from aquaculture, with a value of USD 1.9 million (Table 2) [31]. The main shellfish farming areas are found around Lüderitz, Walvis Bay, and Swakopmund [58]. The Pacific oyster (*C. gigas*) is the primary cultured species in Namibia, known for its relatively shorter growth period of 9 to 15 months compared to competitors in Europe, Japan, and the US [59].

Longlines are the predominant production method in Namibia, although some pond-based production also exists. Oyster (*C. gigas*) seed is provided by a hatchery established in Walvis Bay. The local Namibian market for oysters is limited due to the absence of traditional seafood consumption and the relatively small population of around 2.8 million inhabitants. Growers in Namibia primarily sell to wholesalers, as direct selling to restaurants for small orders requires significant effort [60]. The main markets for Namibian

oysters are South Africa and southeast Asia, with only a small proportion sold in the local market [61]. Namibia is a significant exporter of oyster products to South Africa and a major competitor for local producers there [56]. However, the expansion of the oyster aquaculture industry in Namibia is hindered by limited access to other export markets, such as the EU, the United States, Russia, and Asian countries. Compliance with approved sanitation standards is crucial for Namibian oysters and other high-quality shellfish to enhance access to these export markets [62].

### 3.2. Political, Economic, Social, and Technological (PEST) Considerations in Oyster Aquaculture

3.2.1. Political Considerations (P)

Governance and Licensing

The aquaculture sector faces various challenges related to governance, including legal insecurity, cumbersome administrative procedures, and weak participatory approaches [63]. Good governance and a sound regulatory framework are crucial to address these constraints, along with effective decision-making, planning, and monitoring tools [63]. In this study, licensing requirements for oyster aquaculture were found to vary among the studied markets. For example, the US operates under well-established leasing programs that vary by state, while Brazil has no licensing for on-bottom culture, and Sweden requires culture permits [64]. The duration of leases and permits also differs, with some countries granting 3–5-year approvals (South Africa and Sweden) and France leasing for 30 years [65]. Lease costs are a factor influencing aquaculture development in the US, and high fees may hinder growth and discourage entry into the sector in some states [8].

Moreover, the balance of multiple-use conflicts among maritime actors can cause significant delays and obstacles in permit issuance [66]. Different approaches are employed to address these challenges in the studied markets. As noted above, leasing regulations vary by state, but as one example in the Gulf of Mexico of the US (Alabama), prospective farmers obtain oyster riparian rights either through waterfront property ownership or leasing from other waterfront property owners, ensuring control over designated harvesting areas [67]. In Sweden, fishing laws regulate ownership of natural oyster stocks, with oysters belonging to land/water owners up to 200 m from the shore [52], yet harvest is only allowed in designated production areas for food safety reasons. Leasing rights and culture licensing should also protect ecologically valuable public oyster grounds and consider the natural carrying capacity of bays and inlets, as shellfish production relies on complex ecological processes [52]. In Namibia, licenses are granted based on a positive environmental assessment report, and when no significant environmental risks are identified [58]. Consequently, clear licensing and ownership rights are essential to ensure the sustainability of natural oyster reef exploitation and aquaculture activities. Uncertainty arising from the absence of clear regulations may reduce financial institutions' and investors' willingness to provide necessary funding for expansion in production.

Regulatory Measures Related to Health Considerations

Consuming oysters carries significant health risks due to potential contamination with algal biotoxins, fecal-associated viruses, bacteria, and heavy metals [68]. Consequently, maintaining high food safety standards is essential to protect consumers and ensure the growth of the oyster industry. Hence, to ensure the safety of raw oysters, public health controls and monitoring programs are necessary. The occurrence of such structures appears to be less related to the maturity state of the market than many of the other aspects evaluated in this study. In the United States, the National Shellfish Sanitation Program (NSSP) regulates molluscan shellfish food safety, with oyster-growing waters classified as approved, restricted, or prohibited based on water quality parameters. State shellfish control authorities map areas open or closed for harvesting, and shellfish sanitation programs are funded by permit and license fees [69]. Similarly, the European Union (EU) has stringent regulations for shellfish water classification, with different security measures required based on classifications [70]. EU directives mandate quality standards, health conditions,

and maximum levels of contaminants in shellfish [52]. The European Commission has a National Marine Biotoxin Monitoring Program implemented to detect toxins, and shellfish harvesting and sale are prohibited if unsafe levels are found [71]. The same structures apply to both mature and evolving markets, regardless of developmental stage. In contrast, South Africa and Namibia have no general regulations; however, oyster aquaculture permit conditions prescribe regular testing for heavy metals, biotoxins, and bacterial contamination. Namibia and South Africa have both established shellfish sanitation programs to meet international standards for access to export markets but have not yet achieved EU approval [59]. In Brazil, regulations mainly focus on testing for *E. coli*, toxins (in bivalve meat), and harmful algae concentration, with enforcement challenges in some regions. Brazil has new legislation for the shellfish sanitation program (National Safety Bivalves Mollusks Program MoluBis; Portaria SDA/MAPA 884, 06/09/2023), with standards based on US and EU sanitary legislation, where other contaminants will be included, and area classification will be used. The above shows that the implementation of regulatory measures, such as the Shellfish Sanitation Program and the Marine Biotoxin Monitoring Program across various regions, can potentially contribute to optimizing oyster industry growth at both global and regional levels.

### 3.2.2. Economical Consideration (E)
Challenges and Opportunities for Financing

Securing financing was pointed out as a challenge in oyster aquaculture in all considered markets, except in the US. Small businesses in the industry could benefit from capital investment or low-interest loans provided by provincial and national governments, as recommended in South Africa [57]. In the US, financing options are also available through the Farm Service Administration, aimed at small operators and potential entrants who do not qualify for conventional lending [72]. Moreover, insurance plays a crucial role in ensuring economic security at a larger scale, but it is rare in aquaculture, especially for low-trophic species. The absence of ISO standards for infrastructure dimensioning further complicates the situation. There are, however, examples of how these challenges can be addressed. Aquaculture insurers in North America provide coverage for intensive and semi-intensive production systems, and as such, hatchery and nursery operations qualify for insurance [73]. Nonetheless, the number of insurance policies in effect for oyster crop protection in the US is estimated to be less than 100, covering only 2.5% of the industry [73]. In Brazil, although aquaculture insurance exists for other species, there is demand for this service by the oyster culture. Finding innovative solutions to address financing and insurance gaps is essential for the sustainable growth of the oyster aquaculture sector. In Sweden, the European Maritime, Aquaculture, and Fisheries Fund combined with national funds to implement the national food strategy has been used to channel funding to investments and R&D in the aquaculture sector. In Namibia, financial institutions are assessing ways to provide support to existing shellfish farms, and expertise in aquaculture is being encouraged to assist financial institutions in assessing loan proposals [59].

### 3.2.3. Social Considerations (S)
The Roles of Producer Organizations and Cooperatives

In emerging oyster sectors, the lack of facilitative structures for investment in production-related infrastructure poses challenges for farmers. However, the presence of social networks and producer cooperatives has been found to enhance production and performance within the industry [74]. Small-scale producers benefit from these organizations, as they allow for resource sharing and a unified voice in advocating for the sector's interests. In Ireland, collective efforts in areas such as depuration facilities, food safety standards, logistics, marketing, and insurance against adverse events have been recognized as potential benefits of sector collaboration [45]. Similarly, in Sweden and Brazil, producer organizations and cooperatives provide support in governance, strategic development, day-to-day management, and ensuring food safety standards [41]. The establishment of these collaborative structures

enables information exchange, adaptation to volatile conditions, and the empowerment of oyster growers as primary income sources [74].

3.2.4. Technological Considerations (T)

Seed Production Techniques and Regulation

Extensive oyster seed collection relies on natural variations in abiotic conditions, such as temperature, salinity, and substrate availability. This approach presents challenges in terms of control during the production cycle but offers advantages in robustness and lower resource use [75]. Sea-based seed collection methods, including hand picking, cultch deployment, and collector deployment, are practiced in different regions, indicating a dependence on wild oyster beds (several regions of the United States to varying extents, northern Brazil, eastern South Africa, Sweden, and Ireland). The most extensive collection noted in the studied markets was seed production in Brazil, where most traditional oyster farmers collect naturally settled oyster seed from the roots of the mangrove or purchase such seed from fishermen. Larger oysters are conserved as "mother oysters" to sustain larval production [76]. However, unmanaged seed harvest can deplete natural resources and is therefore sometimes regulated. For instance, Sweden prohibits the harvest of oysters smaller than 6 cm [77], while some US Gulf States restrict seed oyster collection from public reefs to private leases [61]. Louisiana manages public oyster grounds for seed production, allowing oyster producers to obtain seed for cultivation on private leases [78]. In this area, transplanting seeds from public to private areas is considered a restoration activity following natural disasters, as seen after Hurricane Rita and Hurricane Katrina [79]. Cultch deployment is practiced in Ireland and the US but is limited in areas where shallow-water dredging is prohibited, such as Sweden. Transplanting oyster seeds between areas, however, carries significant biosecurity risks, as demonstrated by the spread of oyster herpesvirus (OsHV-1) resulting in mass mortality events in France [41]. Prior to the outbreaks, Marennes-Oleron Bay supplied seed to multiple regions, potentially facilitating the transfer of the virus [80].

Role of Hatcheries in Oyster Seed Production

Hatcheries are vital to oyster aquaculture, serving as a reliable seed source, especially in areas where natural settlement is uncertain or species are locally depleted or extinct, while also aiding genetic improvement and boosting market potential. Namibia's oyster industry exemplifies the advantages of establishing a hatchery. By transitioning from importing oyster seed to domestic production, Namibia achieved self-sufficiency, reduced the risk of importing diseases and invasive species, and developed seeds better adapted to local environmental conditions, resulting in significant production advantages [59]. Hatchery production requires advanced technological and scientific expertise, making it a costly endeavor that benefits from economies of scale [52]. During production, oyster larvae are settled on shell fragments, cultch, or can be produced as a single seed without the addition of cultch. Moreover, recently, larvae have also been produced from remote setting techniques, enhancing the hatcheries' role in oyster production and diversification [81]. Hatcheries can also impact the marketability of oysters by offering desirable traits like shell color patterns and shapes that are heritable traits [82,83]. This will ensure that market-specific attributes are met, for example, the US east coast prefers the lighter shell and mantle colors found in eastern oysters [84].

Hatchery-Produced Seed and the Advantages of Triploid Oysters

Hatcheries play a pivotal role in improving oyster production by selecting broodstock and utilizing genetic advancements. In the Pacific Northwest of the US, British Columbia, Canada, and France, the production of triploid oysters, with one extra set of chromosomes, has revolutionized the oyster industry, as triploids exhibit faster growth rates, consistent quality and texture compared to diploids [85,86]. It is important to note that triploid or tetraploid shellfish are not genetically modified organisms but possess the same genomes

as diploid oysters, with one or two additional sets of chromosomes. Polyploidy can occur naturally in wild oyster populations. Their sterility reduces energy investment in reproduction, allowing for year-round consumption and shorter production cycles. Initially, concerns were raised by producers regarding the consumer acceptability of triploids, but the exceptional performance and quality of triploid oysters have resulted in a booming demand for triploid seed [43]. The development of tetraploid oysters has also greatly facilitated the production of triploid seed, obtained by crossing tetraploid males with diploid females. For example, in the US and France, triploid production has become a significant component of shellfish hatchery output [84]. In order to fully capitalize on the benefits of triploid oyster production and to overcome traditional seasonality perceptions, innovative labelling and consumer education strategies may be necessary. This would shift the perception of triploid oysters as a year-round food item, emphasizing their more consistent quality and flavor [85].

Production Techniques

Oyster cultivation encompasses a range of intensities, from extensive wild harvest-based seed to intensive floating or suspended culture using hatchery-produced seeds. Mature markets have shifted away from relying on the harvest of wild populations while emerging markets still have a mix of wild harvest and aquaculture. Harvest of wild oyster beds results in a product that is less refined than cultivated oysters, showing variations in shape, size, and growth alignments (e.g., cluster oysters)

On-bottom culture farms are particularly susceptible to market price fluctuations and oyster volume sold [87]. Similar practices are seen in one region in Brazil and in Ireland, albeit with different end products. In France, off-bottom culture on trestles has become the predominant technique, accounting for 60% of total production, while suspended culture (5%) involves hanging oysters on ropes or cages in Mediterranean lagoons or open sea lines [81]. In Brazil, various on-growing systems have been developed, including fixed tables for areas with tidal variation, longline suspended systems (the most used system) for deeper waters, and intertidal longlines. In Sweden, most farmers have developed locally adapted versions of different on-growing systems. Deep water methods like longlines can mitigate conflicts near the shore and address capacity issues in bays, allowing for greater production area utilization and expansion in total volume [43].

Postharvest Processing and the Importance of Depuration Facilities

Since adherence to food safety standards is crucial for consumer trust and market expansion [41], various processing methods are employed to enhance the shelf life, quality, and usability of oysters. In addition to traditional methods (i.e., hand sorting and grading), oysters can undergo postharvest processing (PHP) techniques. Depuration plants are the most commonly used PHP method, known for their effectiveness in reducing pathogens while being low-maintenance and free from hazardous byproducts. Other techniques to add value and improve product quality are irradiation, cryogenic individual quick-freezing (IQF), cool pasteurization, and hydrostatic pressure [84]. The establishment of reliable depuration facilities is essential for public health and increasing the value and acceptability of oysters in the market. Brazil has seen the positive impact of partnering with international agencies to establish depuration facilities, resulting in increased sales volume, higher income for growers, and enhanced consumer confidence [41]. Depuration or live storage facilities are also commonly utilized in other regions across the Atlantic, including the United States and Europe (France, Ireland, and Sweden), and in some places in Brazil. The establishment of reliable depuration facilities and setting standards for these facilities may therefore increase the value and acceptability of oysters in the market.

Food Safety through Traceability

Traceability plays a crucial role in ensuring food safety and mitigating risks in the oyster industry. It allows for the tracking of oysters from farm to consumer, particularly

during product recalls, which can have significant health and economic implications [30]. Effective traceability practices contribute to maintaining the reputation of businesses and industries while minimizing negative impacts. In the United States, shellfish tags are vital for traceability, containing essential information such as harvester address, harvest location, and date of harvest. These waterproof tags are required by law to be stored at the final point of sale for 90 days [30]. Additionally, wholesalers play a key role by assigning lot numbers to shipments and maintaining metadata about the product. While digital traceability technology has been developed and piloted in some regions, its widespread adoption remains limited [88].

Similarly, in the European Union, traceability regulations mandate that all fisheries and aquaculture products be traceable throughout the production, processing, and distribution stages. Adequate labelling with essential information, including species identification, production method, catch or farming area, and fishing gear category, is required [89]. These measures ensure transparency and facilitate effective traceback in the event of issues or recalls. Compliance with traceability requirements enhances consumer confidence, safeguards public health, and supports the responsible and sustainable development of the oyster industry.

## 4. Conclusions and Recommendations

Oyster production and consumption across Atlantic regions show marked differences in production methods, species farmed, and market dynamics. The US, France, and Brazil have robust oyster industries, while Namibia and South Africa show growth potential despite challenges. Ecological concerns arise from seed production techniques, though innovations such as hatcheries and triploid oysters offer benefits, despite resource and acceptance issues. Public health measures and a range of postharvest processing methods, focusing on depuration facilities and traceability, are vital for consumer protection and industry growth. To tackle financial hurdles, particularly for small businesses, policy reforms, innovative solutions, and government support are necessary. Enhanced insurance coverage is needed for risk mitigation and industry stability. Regulatory frameworks and property rights need strengthening to foster responsible oyster exploitation and sustainable aquaculture. The establishment of hatcheries is recommended for stable seed supply, marketability, and reducing dependence on imported seeds. Improvement in postharvest processing methods and robust traceability systems are essential for food safety and market value. While continued research is essential to produce innovative practices, enhance cultivation techniques, and ensure responsible market dynamics, this study concludes that to achieve significant growth at both global and regional levels, identified variations may need to be harmonized. While this study produced important findings, it is vital that more comprehensive research on future market dynamics and consumer preferences across regions be conducted for effective market strategies. Additionally, assessing the impact of regulatory changes on the oyster industry's growth and sustainability, as well as exploring advanced risk management strategies and insurance models tailored to the specific needs of the oyster industry, may be crucial steps for future research.

**Author Contributions:** Conceptualization: Å.S., D.F., S.S. and J.A.I.; Methodology: Å.S., D.F., I.E.P., S.S. and J.A.I.; Validation: Å.S., D.F., J.A.I., P.B. (Pierre Boudry) and C.H.; Analysis: D.F., Å.S. and J.A.I.; Investigation, D.F. and Å.S.; Data Curation: Å.S., C.J.B., D.F.; C.H. and J.A.I.; Writing—Original Draft Preparation: D.F., J.A.I. and Å.S.; Writing—Review and Editing: Å.S., D.F., C.J.B., J.A.I., T.A., P.B.(Peter Britz), D.D., C.H., I.E.P., W.W., S.G., M.K., F.L., J.F.A.L., A.P.L., S.S. and A.K.M.; Supervision, Å.S.; Funding Acquisition, Å.S. and S.S. All authors have read and agreed to the published version of the manuscript.

**Funding:** This research was financially supported by the EU-funded project AquaVitae (project number 818173).

**Informed Consent Statement:** Informed consent was obtained from all subjects involved in this study.

**Data Availability Statement:** Data is contained within the article.

**Acknowledgments:** This paper is based on the MSc thesis by Daniel Fitzgerald and works from AquaVitae WP 6.

**Conflicts of Interest:** The authors declare no conflict of interest.

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
