# Peer review of "Comparative Description and Analysis of Oyster Aquaculture in Selected Atlantic Regions: Production, Market Dynamics, and Consumption Patterns"

_fishes, doi:10.3390/fishes8120584_

Round 1
Reviewer 1 Report
Comments and Suggestions for Authors
The manuscript explores the production, market, and consumption of the oyster industry. The research topic has value, but there are still areas for improvement or clarity in the discussion process. It is recommended that clarification be provided. In addition, improving the academic contribution of the manuscript and making practical suggestions are essential. More specific as followings:
1. Research purpose and conclusion suggestions:
The purpose of the research is to solve the problem (This can serve as a crucial resource for understanding and optimizing global oyster production, marketing, and consumption strategies.), and to determine whether the suggested strategies provided in the conclusion are suitable for the purpose. The suggestions can be more specific and improve the manuscript's academic contributions.
2. Selection of research scope:
Manuscript Lines 72-74 mention that China, France, the United States, South Korea, Japan, and the Philippines are the major countries in global oyster production, but the scope of the study and the selection of countries (Line 120-121) are not clearly discussed.
3. Selection and questionnaire of academic experts:
How the academic experts mentioned in Line 140-144 came into being. In addition, questionnaires are open-ended, semi-open-ended, or closed-ended, and how the questionnaire is generated.
4. Research methods:
Can the author explain more clearly how to select the factors for PEST in the manuscript, why financing is the primary consideration in the economy, and social networks and producer cooperatives are social considerations?
Comments on the Quality of English LanguageModerate editing of English language required
Author Response
- Research purpose and conclusion suggestions: The purpose of the research is to solve the problem (This can serve as a crucial resource for understanding and optimizing global oyster production, marketing, and consumption strategies.), and to determine whether the suggested strategies provided in the conclusion are suitable for the purpose. The suggestions can be more specific and improve the manuscript's academic contributions.
Response: We have added a sentence(L118-118) to strengthen the significance of the study. We have also revised the conclusion to provide guidance for future research (L638-643).
- Selection of research scope: Manuscript Lines 72-74 mention that China, France, the United States, South Korea, Japan, and the Philippines are the major countries in global oyster production, but the scope of the study and the selection of countries (Lines 120-121) are not clearly discussed.
Response: To clarify the scope of the study, we have modified the sentence in the method section(L125-127). It now reads: “These regions were selected based on their participation in or association with the AquaVitae project, to provide a snapshot of variability among nations/regions based on overall market trends”.
- Selection and questionnaire of academic experts: How the academic experts mentioned in Lines 140-144 came into being. In addition, questionnaires are open-ended, semi-open-ended, or closed-ended, and how the questionnaire is generated.
Response: In addition to their selection based on substantial knowledge, we have now specified that their selection was also based on their participation in or association with the AquaVitae project(L144). We have also specified the questions were open-ended (L141).
- Research methods: Can the author explain more clearly how to select the factors for PEST in the manuscript, why financing is the primary consideration in the economy, and why social networks and producer cooperatives are social considerations?
Response: We have now added a paragraph in the method section (L154-166) to clarify this. It reads: “For the application of PEST, the political factors (P) account for the political situation. In this study, they included governance, licensing, and regulatory measures related to health considerations. The economic factors (E) cover issues that may influence business operations and profitability. This study included issues related to the challenges and opportunities for financing. Social (S) factors cover aspects related to consumers and producers, with this study incorporating aspects of the roles of producer organizations and cooperatives. The technological (T) factors cover issues related to operations and production, which are critical for the long-term future of oyster farming in various markets. Technological considerations in this study included seed production techniques and regulation, the role of hatcheries in oyster seed production, the advantages of triploid oysters, production techniques, post-harvest processing, the importance of depuration facilities, and food safety through traceability.”
Reviewer 2 Report
Comments and Suggestions for Authors
The title of the article does not correspond with its content since it does not present a comparative analysis, just a superficial description of each market.
The methodology applied in the article is rather poor and insufficient to provide a market analysis. For instance when comparing consumption patterns different methods should be used, e.g. based on a survey. I would expect more advanced methods in an article published in a journal with IF.
The data and results presented should be comparable and not simply described. They could be aggregated in a table or a diagram to make concluding easier.
The authors should provide more developed conclusions, recommendations for market actors and further research.
Keywords are incorrectly elaborated.
There is no clear objective, research questions, or hypothesis.
The paper is rather a review, not an article. It should be thoroughly revised.
Author Response
- The title of the article does not correspond with its content since it does not present a comparative analysis, just a superficial description of each market.
Response: We have modified the title to include the word "description," but have kept the title substantively the same, as the study also included analyses that compared various markets, making the title fitting.
- The methodology applied in the article is rather poor and insufficient to provide a market analysis. For instance when comparing consumption patterns different methods should be used, e.g. based on a survey. I would expect more advanced methods in an article published in a journal with IF.
Response: We disagree with the reviewer; however, we have added more information to the method (L125-127, L41, L44, and L154-166) to make it clearer.
- The data and results presented should be comparable and not simply described. They could be aggregated in a table or a diagram to make concluding easier.
Response: We agree with the reviewer. We have however not created a new table or diagram as the comparable aspects of the market are sufficiently covered and summarized in Table 2
- The authors should provide more developed conclusions, recommendations for market actors and further research.
Response: We have improved the conclusions and recommendations by adding two sentences (L638-643)., which read: “While this study has produced important findings, it is vital that more comprehensive research on future market dynamics and consumer preferences across regions be conducted for effective market strategies. Additionally, assessing the impact of regulatory changes on the oyster industry's growth and sustainability, as well as exploring advanced risk management strategies and insurance models tailored to the specific needs of the oyster industry, may be crucial steps for future research”.
- Keywords are incorrectly elaborated.
Response: We disagree with the reviewer's suggestion and have kept the keywords as they are.
- There is no clear objective, research questions, or hypothesis.
Response: To make the objective and research question clear, we have revised the text, and it now reads(L109-118): “This study, therefore, aimed to conduct a qualitative description of oyster production, consumption patterns, and market dynamics in seven regions (US, Brazil, France, Sweden, Ireland, South Africa, and Namibia) along the Atlantic Ocean, to enhance understanding of these markets. The research questions included understanding how oyster production methods differ among the seven Atlantic regions, the implications of these differences for the industry's sustainability and growth, the unique consumption patterns of oysters in each region, and how market dynamics and strategies vary across these regions. Comparing these similarities and differences among the regions can serve as a crucial resource for understanding and optimizing global and local oyster production, marketing, and consumption strategies”.
- The paper is rather a review, not an article. It should be thoroughly revised.
Response: We disagree with the reviewer and we are convinced this is an article. However, we have reviewed the article according to the specific suggestions of the reviewer as indicated above.
Reviewer 3 Report
Comments and Suggestions for Authors
Dear Author,
I have appreciate your efforts and I agree with structure and contents of your article.
Best regards
Author Response
Thank you. We have improved the manuscript further as highlighted in within.
Round 2
Reviewer 2 Report
Comments and Suggestions for Authors
The Authors have sufficiently improved the paper, therefore it deserves publication.